

# Early language competence, but not general cognitive ability, predicts children's recognition of emotion from facial and vocal cues

Sarah Griffiths[1], Shaun Kok Yew Goh[1,2], Courtenay Fraiser Norbury[1,3] and the SCALES team

[1] Psychology and Language Sciences, University College London, London, United Kingdom
[2] Centre for Research in Child Development, Office of Educational Research, National Institute of Education, Nanyang Technological University, Singapore, Singapore
[3] Department of Special Needs Education, University of Oslo, Oslo, Norway

## ABSTRACT

The ability to accurately identify and label emotions in the self and others is crucial for successful social interactions and good mental health. In the current study we tested the longitudinal relationship between early language skills and recognition of facial and vocal emotion cues in a representative UK population cohort with diverse language and cognitive skills ($N = 369$), including a large sample of children that met criteria for Developmental Language Disorder (DLD, $N = 97$). Language skills, but not non-verbal cognitive ability, at age 5–6 predicted emotion recognition at age 10–12. Children that met the criteria for DLD showed a large deficit in recognition of facial and vocal emotion cues. The results highlight the importance of language in supporting identification of emotions from non-verbal cues. Impairments in emotion identification may be one mechanism by which language disorder in early childhood predisposes children to later adverse social and mental health outcomes.

## INTRODUCTION

Recognition of emotional cues, such as facial and verbal expressions, is an important social skill. It provides us with information about other people's internal emotional states and helps us to interpret and predict their behaviour. Children have typically acquired the vocabulary for basic emotions by 4–6 years of age (*Baron-Cohen et al., 2010*; *Ridgeway, Waters & Kuczaj, 1985*), but accuracy in identifying non-verbal emotional cues continues to improve into late adolescence (*Grosbras, Ross & Belin, 2018*; *Herba & Phillips, 2004*; *Rodger et al., 2015*). Accurate emotion identification has been linked to positive outcomes later in development, including academic success (*Denham et al., 2012*; *Izard et al., 2001*), social integration (*Sette, Spinrad & Baumgartner, 2017*) and good mental health (*Ciarrochi et al., 2003*).

Corresponding author
Sarah Griffiths,
sarah.griffiths@ucl.ac.uk

A critical part of learning to identify emotions is developing emotional concepts that align precisely with the emotional concepts held by other people. The Theory of Constructed Emotion (TCE; *Gendron & Barrett, 2018*) proposes that language is crucial for acquiring nuanced emotional concepts. Verbal labels provide a framework to organise highly variable input from the environment into coherent emotion concepts (*Gendron & Barrett, 2018*; *Lindquist, 2017*). Critically, the TCE suggests that the role of language in supporting emotion recognition goes beyond acquisition of emotion vocabulary. Precise conceptual alignment is achieved through communication with others. If an individual has less opportunity to learn about emotion concepts through language, their conceptual alignment would be compromised, which would lead to less accurate emotion identification. Previous research has shown that parent–child discourse about emotions predicts children's emotion identification accuracy months later (*Dunn et al., 1991*), consistent with the theory that language aids learning about emotions.

In the current study, we test the hypothesis that language supports development of accurate emotion identification by studying a population that has reduced opportunity to learn about emotion concepts through language. Children with Developmental Language Disorder (DLD; previously known as Specific Language Impairment; *Bishop et al., 2017*) have difficulties with receptive and/or expressive language that cannot be explained by a sensory deficit or neurological impairments (*American Psychiatric Association, 2013*). Unlike children with autism spectrum disorders (ASD), children with DLD do not have primary social or emotional difficulties, so any problems with emotion recognition are likely to be a consequence of difficulties acquiring language. If language is necessary for emotion conceptual alignment, children with DLD should have persistent difficulties with emotion understanding, due to reduced opportunity to learn about emotion concepts through language.

Children with DLD have been shown to have difficulty with some aspects of emotion understanding; including identifying emotions from hypothetical scenarios (*Ford & Milosky, 2003*; *Spackman, Fujiki & Brinton, 2006*) and deciding when emotions should be hidden to conform to social display rules (*Brinton et al., 2007*). Some studies have found that school-aged children with DLD have difficulty labelling and categorising facial (*Bakopoulou & Dockrell, 2016*; *Taylor et al., 2015*) and verbal (*Boucher, Lewis & Collis, 2000*; *Taylor et al., 2015*; *Fujiki et al., 2008*) emotional expressions. However, other studies have not found these differences (*Creusere, Alt & Plante, 2004*; *Loukusa et al., 2014*; *Trauner et al., 1993*) or have observed differences only for some emotions (*Spackman et al., 2005*). This equivocal evidence is likely due to variable diagnostic criteria, the heterogeneity of the tasks used, and reduced statistical power due to the small sample sizes. The estimated effect size for emotion recognition deficits in ASD, which we would assume to be larger than the size of any deficit in DLD (due to primary challenges in social-emotional processing), is estimated to be 0.41 (*Uljarevic & Hamilton, 2013*). Power calculation suggests a sample size of 135 participants in each group is needed to reliably detect an effect of this size (*Uljarevic & Hamilton, 2013*). Therefore, much larger studies are required to determine whether children with DLD do have difficulties with emotion identification.

A number of cross-sectional studies in the typically developing population have found associations between language competence and the ability to label and match emotional facial expressions in early childhood (*Beck et al., 2012*; *Pons et al., 2003*; *Rosenqvist et al., 2014*), although other studies have failed to find this relationship (*Herba et al., 2006*; *Herba & Phillips, 2004*). Concurrent relationships between emotion recognition performance and language competence in early childhood may be the result of children not having the vocabulary to meet the language demands in the task. Stronger support for a role of language in refining emotional concepts would come from studies demonstrating a longitudinal relationship between language competence in early childhood and later accuracy in applying labels to emotion cues, at an age when children have acquired basic emotion vocabulary.

In the current study, we use data from a well-characterised longitudinal population cohort that includes children with the full spectrum of language abilities. Children at risk for language disorder were purposefully oversampled, resulting in a cohort that includes a disproportionately large number of children that meet the criteria for DLD. First, we test the hypothesis that early language competence (age 5–6) is associated with the ability to match facial and vocal emotion cues to basic emotion labels in middle childhood (age 10–12) controlling for children's non-verbal cognitive ability. Second, we test the hypothesis that emotion recognition at age 10–12 is poorer in children that met the criteria for DLD at age 5–6, compared to children with typical language. Finally, we look at error patterns to explore whether children with DLD make similar errors on the emotion recognition tasks to their peers with typical language. The analysis plan for this study was preregistered on the Open Science Framework (https://osf.io/pwcms/).

## MATERIAL AND METHODS

### Sample description

Data are taken from the Surrey Communication and Language in Education Study (SCALES). This study has followed a cohort of children who entered state-maintained schools in the county of Surrey in the United Kingdom in September 2011. Language and communication skills were assessed at school entry via a teacher report questionnaire (Children's Communication Checklist-Short; CCC-S; *Norbury et al., 2016*). Based on screening, children were classified as having (1) no phrase speech (NPS) (2) high risk for DLD (3) low risk for DLD. Children were classified NPS if their teacher responded 'no' to the question 'is the child combining words into phrases or sentences?' The CCC-S is not applicable for children not speaking in phrases so these children were given the maximum score. The cut-off between high and low risk status was based on age and sex specific cut-offs on the CCC-S derived from the entire screened population $n = 7,267$) (see *Norbury et al., 2016* for details).

Stratified random sampling identified a subset of 636 children from the screened population who were invited to take part in direct assessments conducted by trained researchers. Exclusion criteria were (1) attending special schools for children with severe intellectual or physical disability and (2) having English as a second language. All remaining

children identified as being NPS ($n = 48$) were invited, as were 233 low risk and 355 high risk children. Sampling all NPS children and oversampling high-risk children ensured that we had sufficient numbers of children that met the criteria for DLD in the cohort. Five hundred and twenty nine monolingual children were assessed in Year 1, and 384 of these were assessed in Year 6. This final assessment included the emotion recognition tasks. Assessments took around two hours and typically took place at school, although a small number took place during home visits.

Language assessments in Year 1 were used to calculate composite scores for expressive language, receptive language, vocabulary, grammar and narrative skills (*Norbury et al., 2016*). Children were classified as meeting criteria for Developmental Language Disorder (DLD) if they scored -1.5 SD below the mean on at least 2 out of 5 of these composite scores in Year 1. This is in-line with DSM-5 criteria for Language Disorder that states children must be substantially and quantifiably below age expectations for language across modalities including vocabulary, sentence structure, and discourse (*American Psychiatric Association, 2013*).

Standard scores from block design and matrix reasoning (WPPSI-III; *Wechsler, 2003*) in Year 1 were used to calculate a non-verbal IQ composite by taking the mean of the two scores. This was used to identify children with suspected intellectual disability, defined as a non-verbal IQ composite score of less than -2 SD below the mean. Children that met DLD criteria in Year 1 were additionally classified as having DLD with no known associated biomedical condition or DLD with a known associated biomedical condition (hereafter termed LD+). Inclusion criteria for 'known associated biomedical condition' were (1) intellectual disability based on non-verbal IQ assessments, and/or (2) teacher reported diagnosis of a biomedical condition. Biomedical conditions included; autism, hearing/visual impairment, Down syndrome, epilepsy, neurological impairment, cerebral pals condition including intellectual disability, autism, hearing/visual impairment, Down syndrome, epilepsy, neurological impairment, cerebral palsy, neurofibromatosis and Noonan syndrome (*Norbury et al., 2016*).

## Consent

Consent procedures and study protocol were developed in consultation with Surrey County Council and approved by the Royal Holloway Ethics Committee (where the study started) in Year 1 and the University College London (UCL) Research Ethics Committee in Year 6 (9733/002). Informed consent was collected from parents before in-depth assessments in Year 1 and Year 6. Informed assent was collected from children prior to the Year 6 assessment. Children were given certificates and small prizes at the end of each assessment session.

## Sample size and power calculations

We conducted *a priori* sensitivity analyses in G-Power based on a sample size estimate of 399 participants (assuming a retention rate of 80% from previous assessment time-point in Year 3). Sensitivity analysis suggested we would have 90% power to detect small ($r = .15$) associations between language and emotion recognition accuracy in the whole sample

(*Cohen, 2013*). We also conducted a sensitivity analysis for assessing the group difference between DLD group and the rest of the sample. Assuming equal attrition we estimated that we would have 103 children in Year 6 that had met the DLD criteria in Year 1, including 70 with DLD and no additional diagnosis. Sensitivity analysis suggested that this would provide 90% power to detect small-medium group differences ($d = 0.34$ for comparison with full DLD group and $d = 0.38$ for comparison excluding children with LD with known origin) (*Cohen, 2013*).

## Assessment procedure

### Year 1 Language

In Year 1, children completed 6 tasks to assess receptive and expressive vocabulary, grammar and narrative language skills. Receptive and expressive vocabulary was assessed using the Receptive/Expressive One word Picture Vocabulary Test (R/EOWPVT-4; *Martin & Brownell, 2010*; *Martin & Brownell, 2011*). These tests have excellent internal consistency for ages 5- to 8-years (Cronbach's $\alpha = .94–.97$) and high test-retest reliability (coefficients $= 0.97–0.98$ for raw scores). Receptive grammar was assessed using a short form of the Test of Reception of Grammar (TROG-S; *Bishop, 2003a*; *Bishop, 2003b*). The manual reports a split-half reliability for the TROG-2 of 0.88, suggesting good internal consistency (*Bishop, 2003a*; *Bishop, 2003b*). Pilot testing demonstrated excellent agreement between short and long forms of $r(17) = 0.88$. Expressive grammar was assessed using the School-Age Sentence Imitation Test (SASIT E32; *Marinis et al., 2011*). Expressive narrative skill was assessed using the narrative recall subtest from the Assessment of Comprehension and Expression 6–11 (ACE 6-11; *Adams et al., 2001*). Cronbach's Alpha of narrative recall for children aged 6- to 11-years is 0.73. Finally, receptive narrative skill was assessed using bespoke questions derived from the ACE 6-11 narrative (*Adams et al., 2001*).

Scores for each test were standardised using the LMS method and then averaged to create composite scores for vocabulary (EOWPVT-4 and ROWPVT-4), grammar (TROG-S and SASIT), narrative (ACE recall and comprehension), receptive language (ROWPVT-4, TROG-S and ACE comprehension) and expressive language (EOWPVT-4, SASIT and ACE recall). Scores on these 5 language composites were used for diagnosing DLD (see participant section above). A total language composite score was created by averaging the standard scores for all 6 tests. This language composite was used in the analysis.

### Year 1 Non-verbal IQ assessment

In Year 1 children completed two tests of non-verbal IQ (NVIQ); (1) Wechsler Preschool and Primary Scale of Intelligence 3rd edition Block Design and (2) Matrix Reasoning subtests (WPPSI-III; Wechsler, 2003). Standard scores on these two tasks were averaged to create a NVIQ composite score (*Norbury et al., 2016*).

### Year 6 Emotion recognition

In Year 6 children completed two emotion recognition tasks; one to measure recognition of emotion from faces and one to measure emotion recognition from voices. Each task consisted of 60 trials in which children were presented with photos of faces or recordings of vocal sounds corresponding to one of 6 emotions (happy, sad, angry, surprised, scared and

disgusted). For the facial expression task, stimuli were photos of 10 adult actors (5 female and 5 male) selected from the Radboud Faces Database (*Langner et al., 2010*). For the vocal expression task, non-verbal sound stimuli were selected from a validated set of emotional vocal sounds (*Sauter et al., 2010*) that have previously been used in research with 6–10 year old children (*Sauter, Panattoni & Happé, 2013*) and adults with autism (*Jones et al., 2011*). The sounds are made by 4 adult actors (2 male and 2 female).

In both tasks, participants were shown a fixation cross for 500 ms, followed by the face stimuli for 2 s, or the audio clip accompanied by a cartoon image of a listening man. Participants were then presented with 6 buttons with the emotion labels in a circular formation on the screen. The labels remained until the participant made a response by pressing the button on the touch screen. The order of the emotion labels on the screen was randomised between participants and tasks, but kept the same between trials for each participant. Total accuracy scores were calculated out of 60 for each task separately.

Before completing the task, we checked children's understanding of the 6 emotion words by asking them to read the labels aloud and describe or imitate that emotion. If the child was unable to describe or imitate one or more of the emotions, the assessment was terminated as it was assumed they did not have the basic emotion vocabulary. A very small number of children were not able to read the labels but could describe or imitate the emotion when the word was said aloud. For these children the researcher asked them to give their response verbally during the task and entered their responses for them.

### Analysis plan
#### Standardisation of scores
Test scores from each of the six language assessments and the two NVIQ assessments in Year 1 were standardised using the LMS method (*Vamvakas et al., 2019*). LMS is a method of standardisation based on the Box–Cox transformation that converts scale raw scores to normality. The resulting scores reflect standardised scores adjusted for age, with a mean of 0 and a standard deviation of 1. We planned to standardise emotion recognition scores using the same method but this was not necessary as performance was not correlated with age in our sample (faces $r = .05$, $p = .37$; voices $r = .002$, $p = .97$).

#### Sampling weights and missing data
Sampling weights were included in all analyses to account for study design and any bias in attrition. This adjustment means that estimates are representative of the screened sample of 6,459 monolingual children in state-maintained schools. Sampling weights were produced by multiplying the inverse of the predicted probability of two logistic regression models that predict inclusion in the sample. The first regression model estimates a child's likelihood of being initially invited into the study. This was fitted to the entire population of 6,459 monolingual children in mainstream schools that were screened at school entry. The covariates in this model are those that determined selection into the study due to the stratified sampling method. These are total number of children assessed per school, and whether a child was identified as at risk for DLD based on CCC-S teacher ratings (86th centile or above for sex and age group). The second regression model was fitted to the 636 children invited into the study. This model used all available variables to predict

retention. This included individual characteristics such as sex, income deprivation score, special education needs, free school meals, English as additional language, Children's Communication Checklist 2 score (*Bishop, 2003a*; *Bishop, 2003b*), language in Year 1, season of birth, Strengths and Difficulties Questionnaire (*Goodman, 1997*) total difficulties score, and school characteristics such as number of pupils on role, percentage of girls, percentage with SEN, and percentage with free school meals. These variables were tested in a stepwise elimination process and included in the model if they predicted inclusion above a cut-off point of .2.

### *Statistical analysis*

Statistical analyses were conducted in R version 3.5.3 and M-Plus. Structural Equation Models (SEM) were built under robust maximum likelihood estimator which is robust to deviations from normality. To test the hypothesis that language competence in Year 1 predicts emotion recognition from faces and voices in Year 6, path analysis was used to model the association between children's composite language scores in Year 1 and their scores on the facial expression and vocal expressions tasks in Year 6. Additionally, because one previous study had suggested that children with DLD may be more impaired in recognition of emotion cues from voices rather than faces (*Trauner et al., 1993*) we compared the strength of the pathways between language and performance on the facial expression task and vocal expression task using Wald test of parameter constraints. Finally, we then entered Year 1 NVIQ composite into the model to assess whether language scores continue to predict emotion recognition after accounting for variation in non-verbal cognitive ability.

We then compared children in the DLD group to the rest of the sample on total accuracy from the emotion recognition tasks separately. We did not control for NVIQ in this analysis because low NVIQ is not an exclusion criterion for DLD (*Bishop et al., 2017*) and language severity is associated with NVIQ (*Norbury et al., 2017*). This means 'controlling' for group differences in NVIQ would 'control' for relevant and non-random differences between the two groups (*Dennis et al., 2009*). We conducted this analysis both with and without removing children with additional diagnoses, to determine if there was still a group difference after removing children with co-occurring conditions that have also been associated with problems with emotion recognition (e.g., autism and/or severe intellectual disability).

## RESULTS

Of the 384 participants who were seen for assessment in Year 6, 362 (including 67 with DLD and 29 with LD+ additional diagnoses) completed the facial emotion recognition task and 359 (63 with DLD and 27 with LD+ additional diagnoses) completed the vocal emotion recognition task. Three hundred and sixty nine completed at least one task (67 with DLD and 30 with LD+ additional diagnoses) so were included in the analysis. Of the 15 children that did not complete either task, six met criteria for language disorder in Year 1. These children did not complete the task because they did not have the basic emotion vocabulary or were otherwise unable to engage in the task. The other nine children did

**Table 1 Descriptive statistics for the full sample and each language group separately.** The language composite score is the average of standard scores from the six language assessments. The NVIQ composite is the averaged of standard scores from the two nonverbal IQ assessments. Emotion recognition scores are raw total accuracy scores on each task.

| | Full sample $n = 369$ Mean (SD) | Typical language $n = 272$ Mean (SD) | DLD $n = 67$ Mean (SD) | LD+ $n = 30$ Mean (SD) |
|---|---|---|---|---|
| Age Year 1 (Years) | 5.97 (0.39) | 5.97 (0.40) | 5.90 (0.35) | 6.13 (0.34) |
| Age Year 6 (Years) | 11.16 (0.34) | 11.16 (0.34) | 11.12 (0.36) | 11.24 (0.31) |
| Male n (%) | 185 (50%) | 129 (47%) | 37 (55%) | 19 (63%) |
| Language composite Year 1 | −0.59 (1.06) | −0.09 (0.85) | −1.78 (0.49) | −2.50 (0.67) |
| NVIQ composite Year 1 | −0.39 (1.07) | −0.13 (0.98) | −0.83 (0.65) | −1.81 (1.19) |
| ER faces Year 6[a] | 0.76 (0.12) | 0.79 (0.10) | 0.71 (0.14) | 0.61 (0.13) |
| ER voices Year 6[b] | 0.75 (0.11) | 0.78 (0.08) | 0.71 (0.12) | 0.59 (0.16) |

Notes.

NVIQ, non-verbal IQ; ER, Emotion recognition.

[a] Based on 362 total, 272 TL, 67 DLD, 29 LD+

[b] Based on 359 total, 278 TL, 63 DLD, 30 LD+

not have DLD and did not complete the tasks due to technical issues. Table 1 provides descriptive statistics for all variables for the total sample, and DLD, LD+, and typical language groups separately.

Attrition was slightly higher than we had anticipated when we conducted our *a priori* sensitivity analysis. However, our achieved sample size still gave us 90% power to detect small associations between language and emotion recognition ($r = 0.15$). We also still had 90% power to detect small-medium size group differences in emotion recognition accuracy between the DLD group and typical language group ($d = 0.35$), even after excluding those with LD+ additional diagnoses ($d = 0.41$).

## Does early language competence predict later emotion recognition accuracy?

There were moderate prospective relationships between language competence in Year 1 and emotion recognition from vocal expressions ($\beta = .40$, S.E $= .06$, 95% CI [.28–.51]) and facial expressions ($\beta = .42$, S.E $= .06$, 95% CI [.30–.55]) in Year 6. Wald's test of parameter constraints did not provide evidence for a difference in path strengths between language and emotion recognition from faces and language and emotion recognition from voices ($X^2(1) = 2.51$, $p = .11$). We had planned on combining the two emotion recognition scores into a single composite score if there was no evidence for a difference in path strengths. However, the correlation between the two outcomes estimated in the model was not sufficient to justify this ($r = .37$, S.E $= .06$, 95% CI [.25–.50]).

When NVIQ in Year 1 was entered into the path model as a predictor, the relationships between language and performance on the two emotion recognition tasks remained, and there was no statistical evidence for a prospective relationship between NVIQ and performance on either emotion recognition task (see Fig. 1 for the standardised regression coefficients and confidence intervals for these paths). Table 2 provides the correlation matrix for the variables in the model.

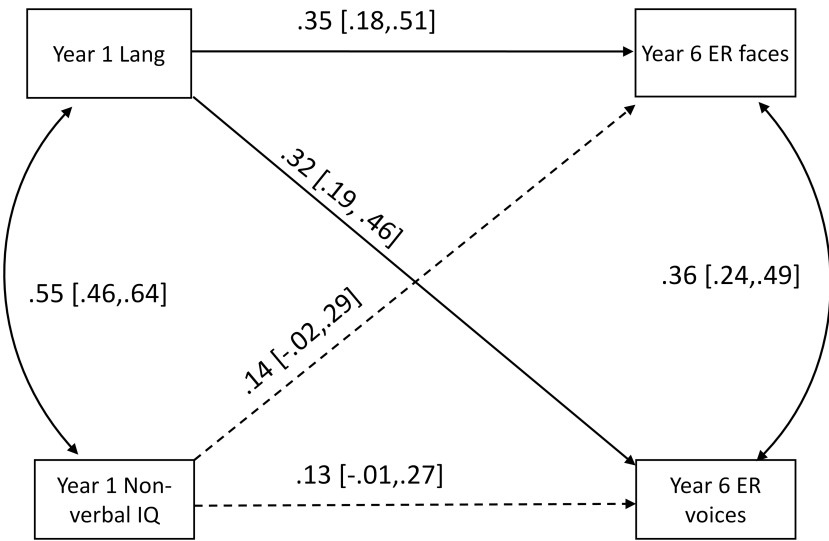

**Figure 1  Path model showing prospective relationships from language (Lang) and non-verbal IQ (NVIQ) in Year 1 to emotion recognition from faces (ER faces) and voices (ER voices) in Year 6.** Significant paths are solid lines while insignificant paths are dashed lines.

**Table 2  Correlations between variables included in the path model.** The language composite score is the average of standard scores from the 6 language assessments. The NVIQ composite is the averaged of standard scores from the 2 non-verbal IQ assessments. Emotion recognition scores are raw total accuracy scores on each task.

|  | Language composite Year 1 | NVIQ composite Year 1 | ER faces Year 6 |
|---|---|---|---|
| NVIQ composite Year 1 | 0.55 |  |  |
| ER faces Year 6 | 0.42 | 0.33 |  |
| ER voices Year 6 | 0.40 | 0.31 | 0.48 |

Notes.

NVIQ, non-verbal IQ;  ER, Emotion recognition.

## Do children with DLD have poorer emotion recognition skills than their peers with typical language?

Figure 2 illustrates the distributions of raw accuracy scores on the facial and vocal emotion recognition task for children diagnosed with DLD and children with language in the typical range. Weighted t-tests provided clear statistical evidence for a large group difference in recognition of emotions from faces; $t(360) = 4.06$, $p < .001$, $d = .90$, and voices $t(353) = 4.24$, $p < .001$, $d = 0.89$, between these groups. When children with LD+ additional diagnoses were removed, the effect sizes reduced slightly but there was still evidence for a medium–large group difference for recognition of emotion from faces; $t(331) = 2.72$, $p = .007$, $d = .72$, and voices; $t(326) = 2.87$, $p < .001$, $d = .78$.
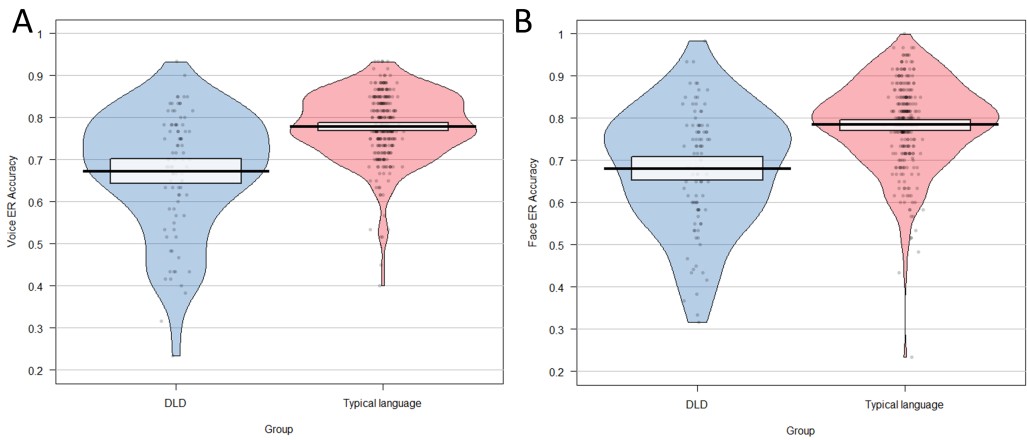

**Figure 2** Pirate plot showing distribution of total scores on (A) the vocal emotion recognition task and (B) the facial emotion recognition task for group with DLD and the typically language group.

## Do children with DLD make similar errors in emotion recognition tasks to their peers with typical language?

In order to explore possible differences in the kinds of errors made by children with DLD and those without DLD, we created confusion matrices for each task for each group of children (Fig. 3). From these it can be seen that in general the pattern of errors is very similar across the two groups. The most commonly misidentified emotion in the facial emotion recognition task was disgust in the DLD group and fear in the typical language group and the least commonly misidentified emotion was happiness in both groups. For the vocal emotion recognition task, the most commonly misidentified emotion in both groups was surprise and the least commonly misidentified emotion was happiness in the DLD group and disgust in the typical language group.

## DISCUSSION

In the present study we examined the prospective relationship between language competence in early childhood and identification of non-verbal emotion cues in middle childhood in a large population-derived cohort of children with diverse language and cognitive skills. We found evidence for a moderate positive association between language competence at age 5–6 and recognition of facial and vocal emotional cues at age 10–12 supporting our hypothesis that early language skills are positively associated with later emotion recognition ability. The relationship between early language and later emotion recognition held when adjusting for non-verbal cognitive ability, suggesting it is language specifically, rather than cognitive ability more generally, that is associated with later emotion recognition ability.

This is the first longitudinal evidence to support the hypothesis from the TCE that language competence plays a role in supporting accurate identification of non-verbal emotion cues (*Gendron & Barrett, 2018*). While previous studies have found concurrent associations between language and emotion recognition ability (*Beck et al., 2012*;
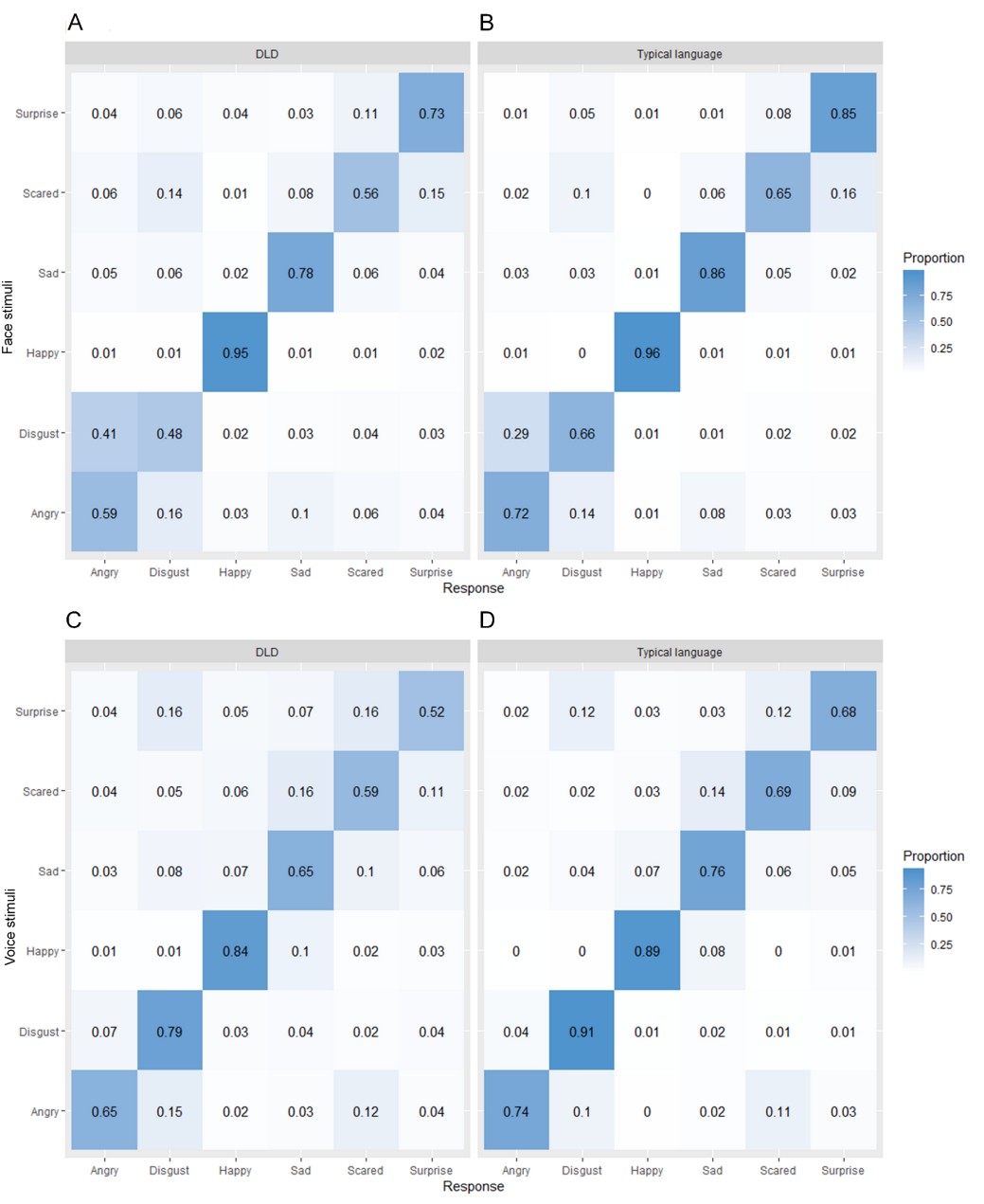

**Figure 3** Confusion matrices showing proportion of responses in each category for each presented emotion separately by language group for (A, B) faces and (C, D) voices.

*Rosenqvist et al., 2014*), concurrent associations may be explained by children's language skills limiting their ability to engage with the task. The longitudinal association identified in this study is consistent with the hypothesis derived from the TCE that having poor language skills has a longer term impact on children's emotion recognition abilities due to children having less refined emotional concepts.

We also found that children with DLD have a large deficit in emotion recognition ability. These results help clarify contradictory literature on whether children with DLD have deficits in emotion recognition (*Bakopoulou & Dockrell, 2016*; *Boucher, Lewis & Collis, 2000*; *Creusere, Alt & Plante, 2004*; *Loukusa et al., 2014*; *Taylor et al., 2015*; *Trauner et al., 1993*). Many of the previous studies have been small ($n < 20$ children with DLD; *Boucher, Lewis & Collis, 2000*; *Loukusa et al., 2014*; *Taylor et al., 2015*; *Trauner et al., 1993*) and have therefore lacked the statistical power to detect the expected medium-small effect size (*Uljarevic & Hamilton, 2013*). Ninety-seven children in our cohort that met the DSM criteria for language disorder when they were 5–6 years old (*American Psychiatric Association, 2013*), based on rigorous linguistic and cognitive testing, completed at least one emotion recognition task in Year 6, giving us sufficient statistical power. We found strong evidence for a large difference in emotion recognition ability at age 10–12 between those that met the criteria for DLD at age 5–6 compared to those with language in the typical range at age 5–6. When children with other diagnoses that associate with both language and emotion recognition difficulties were excluded from the DLD group (e.g., autism or intellectual disability), the group difference attenuated somewhat, but the statistical evidence for differences between children with DLD and their peers remained strong. This is the largest study to compare children with DLD to typical developing peers on emotion recognition performance, providing the best evidence to date for emotion recognition deficits in DLD.

Emotion recognition ability improves with age up until adolescence (*Herba & Phillips, 2004*), so we had expected to find an association between emotion recognition performance and age in this study. The fact that we did not find evidence for age differences in emotion recognition is likely due to the narrow age range in this study (10–12 years) and oversampling of children with suspected DLD. Age related differences within this narrow age range are likely to be small, and therefore obscured by larger individual differences in emotion recognition associated with language disorder.

Our emotion recognition task was verbal in the sense that children had to match non-verbal emotion cues to verbal labels. It could therefore be argued that the verbal demands of the task explain the relationship between language competence and emotion recognition performance. However, the labels were basic emotion words that are highly frequent and well within the vocabulary range of children aged 10–12 (*Baron-Cohen et al., 2010*), even the vast majority of those with DLD. We checked that children understood the emotion words before completing the assessment. A very small number of children with DLD ($n = 6$) lacked the vocabulary to engage in the task, so were not included in the study. Non-verbal tasks, such as a facial expression matching task (*Taylor et al., 2015*), can be completed using visual features alone, without any comprehension of the underlying emotion and so do not truly test emotion identification. We argue that performance on an emotion labelling task is associated with early language competence not just because it involves a verbal label, but because language is involved in developing nuanced emotion concepts through communication with others throughout childhood (*Gendron & Barrett, 2018*).

The ability to recognise and label emotions in the self and others is an important component of social problem solving. The finding that this ability is compromised in DLD may explain why children with DLD are at increased risk of internalising, externalising and ADHD symptoms (*Yew & O'Kearney, 2013*). The causal pathway between DLD and poor mental health outcomes is unclear, but one possibility is that language problems interfere with aspects of social-emotional processing (such as emotion concept development), which in turn leads to negative social, emotional and mental health outcomes. *Im-Bolter, Cohen & Farnia (2013)* found that adolescents referred to mental health services had poorer structural and figural language than peers recruited from the community and were poorer at social problem solving. The findings in the current study raise the possibility that emotion identification may be one pathway in which poor language in early childhood compromises social functioning and mental health in children with and without DLD.

There are a number of possible pathways from poor early language to later emotion recognition difficulties, which should be explored in future studies. First, we assume that children with DLD have less opportunity to learn about emotion through communication with caregivers but we have not directly tested this. Future studies should explore the quality of parent–child discourse about emotions and test whether this is associated with children's emotion recognition. A second promising avenue for future research is the role of alexithymia in explaining the association between language and emotion recognition. Alexithymia describes difficulties identifying and reporting one's own emotional state. It has been proposed recently that language disorder is one route to alexithymia (*Hobson et al., 2019*) and alexithymia is associated with impairments in recognising non-verbal emotional cues in others such as facial expressions (*Cook et al., 2013*). Future studies should include measures of alexithymia to determine to what extent this trait explains the association between early language and later emotion recognition ability.

Although our findings are consistent with a causal relationship between language competence and emotion recognition, they cannot provide proof of causality. One way to investigate whether this relationship is truly causal would be to test whether interventions aimed at improving language have positive, cascading effects on emotion recognition skills later in development. Interventions that focus specifically on language skills directly related to emotion understanding are more likely to transfer on emotion recognition than general language interventions, which may be too distal for transfer to occur. To date there has been one preliminary study ($n = 208$) investigating whether a nine-week intervention focusing specifically on improving language related to emotion through storybooks improves other emotional skills in typically developing 7–9 year old children. The intervention group showed improvements in emotional vocabulary, emotion knowledge and recognition of masked emotions from vignettes compared to a 'treatment as usual' control group straight after the intervention ($\beta = 1.05\text{--}1.32$; *Kumschick et al., 2014*). Future research is needed to determine whether interventions focused on language for emotion can improve emotion recognition skills in children with DLD.

## CONCLUSIONS

In conclusion, this study provides the first longitudinal evidence that early language skills specifically predict later emotion recognition from both facial and vocal cues. These findings support the hypothesis that language plays a role in supporting emotion identification (*Barrett, Lindquist & Gendron, 2007*; *Gendron & Barrett, 2018*; *Lindquist, 2017*). Children with DLD are therefore especially vulnerable to difficulties recognising their own and others' emotional states. We propose that this deficit may be one causal mechanism that underpins the reported relationship between early language skills and later adverse mental health.

## ACKNOWLEDGEMENTS

We thank Surrey County Council for facilitating the data collection process and the children, parents, schools and teachers for taking part in the study. We also thank the other members of the SCALES team: Debbie Gooch, Gillian Baird, Tony Charman, Andrew Pickles and Emily Simonoff for their advice. Finally, we thank Dorothy Bishop for permission to develop the Children's Communication Checklist-Short and allowing us access to the standardization data. The views expressed in this article are those of the authors and not necessarily those of the Wellcome Trust, the ESRC, the British Academy or Surrey County Council.

### Funding

This research was supported by grants from the ESRC (ES/R003041/1) and the Wellcome Trust (WT094836AIA) to Courtenay F. Norbury, and a British Academy Visiting Fellowship awarded to Shaun K.Y. Goh (VF1/101436). The funders had no role in study design, data collection and analysis, decision to publish, or preparation of the manuscript.

### Grant Disclosures

The following grant information was disclosed by the authors:
ESRC: ES/R003041/1.
Wellcome Trust: WT094836AIA.
British Academy Visiting Fellowship: VF1/101436.

### Competing Interests

The authors declare there are no competing interests.

### Author Contributions

- Sarah Griffiths conceived and designed the experiments, performed the experiments, analyzed the data, prepared figures and/or tables, authored or reviewed drafts of the paper, and approved the final draft.
- Shaun Kok Yew Goh and Courtenay Norbury conceived and designed the experiments, authored or reviewed drafts of the paper, and approved the final draft.

## Human Ethics

The following information was supplied relating to ethical approvals (i.e., approving body and any reference numbers):

Consent procedures and study protocol were developed in consultation with Surrey County Council and approved by the Royal Holloway Ethics Committee (in where the study started) in Year 1 and the UCL Research Ethics Committee in Year 6 (9733/002).

## Data Availability

Materials, data and code are available at the Open Science Framework: Griffiths et al. (2018): ''Early language difficulties and emotion recognition in middle childhood: Evidence from the SCALES cohort''. OSF. dataset. https://osf.io/8c4v7/.

Twelve participants did not consent to data sharing so their data has been removed from the dataset shared at OSF. The results obtained from running the code on the data provided will therefore differ slightly from those found in the uploaded output and article.

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
