# Peer review of "Early language competence, but not general cognitive ability, predicts children’s recognition of emotion from facial and vocal cues"

_PeerJ, doi:10.7717/peerj.9118_

## Round 0.1 · original submission · Minor Revisions

I have now received three reviews of your manuscript, and I would like to thank all reviewers for their timely and valuable comments.

All reviewers have provided comprehensive feedback on the manuscript, and their reviews are appended below. I will not reiterate the comments here other than to note that in general the required revisions are relatively minor.

Please ensure you address all reviewer comments; however, I believe that the following issues warrant particular attention when revising your manuscript:

1) Both Reviewer 1 and Reviewer 2 had queries regarding standardisation of scores and the scoring of the composite language score. These should be clarified in the revised manuscript.

2) Reviewer 2 suggested elaborating on other factors that might contribute to emotion recognition skill. Related to this, Reviewer 3 suggested incorporating some discussion of links between alexithymia and language difficulties. This is an excellent suggestion, which I think would also speak to Reviewer 2's comment regarding elaborating on other relevant factors.

·

Basic reporting

This manuscript tackles the association between language difficulties and emotion recognition using an existing longitudinal dataset. The introduction provides a concise and relevant background. The following comments may be useful for the authors to consider:

Introduction

- Line56: One hypothesis has been posed – “In the current study, we test the hypothesis that language supports development of accurate emotion identification by studying a population that have reduced opportunity to learn about emotion concepts through language”. However, the results speak to three key questions:
1. Does early language competence predict later emotion recognition?
2. Do children with DLD have poorer emotion recognition skills than their peers with typical language?
3. Do children with DLD make similar errors in emotion recognition tasks to their peers with typical language?

It may be useful to the reader to include more specific hypotheses in the introduction?

- Have there been any systematic reviews or meta-analyses on the association? This would shed light on the reliability of this potential association.

Article structure
- The manuscript was well structured and easy to follow
- Methods section could benefit from additional sub-headings for inclusion and exclusion criteria (see comments below)

Table 1
- Is there a reason for reporting age in “months”? I think this makes sense for young children (< 4 y/o), but is harder to conceptualise for older children. Perhaps consider reporting age in years?
- May be more common to report N for sex rather than percentage
- The test scores reported in the table do not match the test descriptors included in the methods section – what is the language composite?

Experimental design

This research is within the Aims and Scope of PeerJ. It tackles an interesting research question and provides a step towards enhancing our body of knowledge on the relationship between language difficulties and emotional processing. However, I have the following minor comments that could improve the overall quality of the paper:

Materials and Methods
Sample description
- It would be helpful to have more information about the DLD group earlier – rather than in the results
- The NPS group took me by surprise, and I was unsure of the relevance of this group in relation to DND. Perhaps justify this further and earlier?
- Perhaps include the cut off criteria for the CCC-S?
- Line 114: consider re-warding “in in-depth”
- The inclusion and exclusion criteria need to be clearer. Perhaps consider separate subheadings to delineate these criteria?
- Line 119: Can you include details on length of the assessment, location, and how the assessment was conducted?
- Clarify the diagnostic criteria used to classify DLD children – especially considering that this is a perceived strength of the study.
- It would be helpful to clarify the criteria used to classify intellectual disability
- Line 129: state the additional “associated conditions” other than autism

Consent
- Line 133: Rephrase the (in where the study started)
- Line 134: UCL – not previously defined

Assessment procedure
- It would be helpful to have information on the skills that are being measured rather than the tests that are used. For example, what subtests from the ACE are being used, and what language skill does this measure? It would also be helpful to include more information on the reliability and validity of the tests (given the heterogeneity in assessing language difficulties) as well as state the cut-off scores used to indicate DLD.
- Line 154: reference for SASOT E32 is needed
- In the discussion, you refer to DSM diagnostic criteria for DLD. However, this criteria is not used in your methods section to justify group sections. I would recommend integrating DSM criteria into your selection criteria, or removing this section of the discussion

Year 6 Emotion regulation task
- This section was difficult to follow and the expression of this paragraph could be improved – particularly Lines 172 – 178. Could you present the paradigm and/or stimuli visually?
- Line 170: Is there evidence for this task being used with children?
- Line 171: What type of developmental disorders has this task been used in with adults (Jones et al., 2011).

Sampling weights and missing data
- You propose a model that might predict inclusion of the sample. Can the authors please justify the proposed variables? Definitions are also needed for certain variables – CCC-2 score is included but what is this a measure of? Is it the Castles and Coltheart Reading Test? And if so, why would reading predict inclusion in the sample? As is SDQ – strengths and difficulties questionnaire? Why would internalising and externalising difficulties predict sample of included participants?

Validity of the findings

The use of an existing longitudinal database to analyse the association between DLD and emotion recognition adds to our existing knowledge base of this association. The data has been provided and the study has been pre-registered. The following comments could also be considered:
- Remove interpretation of findings from the results section to the discussion
- Is there any data on emotion recognition in Year 1? Or language at Year 6? This would be useful information in terms of the direction of association.
- Consider investigating association between language and poor mental health (using SDQ data) if available – this is mentioned in your discussion and would be interesting to investigate – although perhaps not for the purpose of this manuscript!

Discussion:
- Line 300: Evidence for a positive or negative moderate relationship? Relationship seems to imply causation (to me) – perhaps “association” would be a more accurate term?
- The discussion would be strengthened with further critique and comparison of the findings against previous literature.
- Line 311-312: Not entirely true regarding N < 20 sample size of previous literature. perhaps reference the studies that support your argument?
- Line 313-314: Specific reference to DSM criteria for DLD that has not been mentioned as selection criteria. Please address this point
- Line 322: What makes this the “best” evidence? This line needs work – particularly in regard to the claim that emotion recognition deficits are “directly linked to language proficiency” – the results do not speak to direct links. The authors should be more tentative with this conclusion.
- To add weight to argument that poor language is associated with poor emotion recognition and poor mental health – you could look at associations with emotional health problems as you included SDQ as a variable?

Additional comments

Thank you for the opportunity to review this manuscript. I wish you all the best for your future endeavours!

Reviewer 2 ·

Basic reporting

No comment

Experimental design

The research question is relevant and meaningful. The method and design are clear, appropriate, and well described. I believe that in the emotion recognition task, the facial expressions were of adults and adults produced the non-verbal sound stimuli. I think it would be helpful for the authors to explicitly indicate this.

There is one thing I was a bit puzzled about. In the Assessment procedure section, Year 1 language, the authors indicated that they created a single language composite score by averaging the Z-scores. In the Analysis plan, standardization of scores section, the authors note that they standardized the six language assessments and the two NVIQ assessments using the LMS method. Can the authors clarify how the Z-scores are related to the standardization?

Validity of the findings

The findings represent an important addition to the literature and are clearly reported. I think it would provide additional clarity, however, if the correlations among the variables included in the path analyses were reported. The authors provide the correlation between the two outcome variables but nothing else. I think Table 1 would be improved if the Full sample descriptives were removed since the language groups are of primary interest.

The Discussion provides a well-synthesized interpretation of the findings within the context of previous literature. The authors acknowledge study limitations; however, I think the authors could also acknowledge that there might be other factors that contribute to emotion recognition skill (e.g., mother-child discourse about emotions) that were not included in their study.

·

Basic reporting

This paper is concerned with an interesting aspect of DLD and its implications for the social-emotional well-being of affected children. The paper is generally well-written (see some copy-editing issues below) and describes a carefully-executed study that will be of interest to DLD researchers and clinicians alike. My comments and suggestions below are offered as ways of further strengthening the MS, particularly in relation to placing it in a slightly broader context of work that has gone before in the emotion recognition space.
1. In the introduction/lit review, I suggest providing coverage of the work of Brinton and Fujiki and colleagues, as their work is important to the question of emotion recognition in children with language difficultes, for example this paper:
Brinton, B., Spackman, M. P., Fujiki, M., & Ricks, J. (2007). What should Chris say? The ability of children with specific language impairment to recognize the need to dissemble emotions in social situations. Journal of Speech, Language, and Hearing Research 50(3), 798-811.
2. There is also a body of literature on alexithymia which seems relevant to bring in either in the introduction or the discussion (or both). While I hesitate to reference my own work in reviews, my colleagues and I described high rates of alexithymia in a sample of young offenders a few years ago. Although we did not find its presence correlated with language profiles, we did report it as a significant comorbidity with LD and this seems important in relation to the links between language skills and mental health problems in vulnerable children and adolescents. It is definitely relevant to the provision of support services to children and adolescents with LD. The citation for this work is as follows:
Snow, P.C., Woodward, M., Mathis, M. & Powell, M.B. (2015). Language functioning, mental health and alexithymia in incarcerated young offenders. International Journal of Speech-Language Pathology, 18(1), 20-31. DOI: 10.3109/17549507.2015.1081291.
3. In the initial description of the current study, p. 3 of intro, line 93, reference is made to “…a large number of children with DLD” however it is not possible to discern here what is meant by the term “large” – does this mean for example, that it was an unusually disproportionate subgroup?

Experimental design

The design is appropriate to the questions examined.
In the Method, under the heading Year 6 emotion recognition - line 175 – it is not clear what the nature of the response was that was required by children, but it is implied further down on the same page (line 184) that it was not verbal. This requires clarification.

Validity of the findings

Discussion
a. It would be helpful to comment on the implications, if any of the findings for the Theory of Constructed Emotion that was introduced in the opening to the paper.
b. As noted above, I see clear links in the findings of this study to the literature on alexithymia and would really like to see that brought in to the Discussion. This construct also has implications for the ability of children with LD to engage in counselling, which often has a focus on identifying, discussing and re-framing affective distress – all of which of course is done verbally.
c. While I agree with the possibility that interventions aimed at improving language skills MAY result in improvements in emotional recognition, I think we need to be careful not to over-burden the intervention field too much – it is difficult enough to establish efficacy and effectiveness on the proximal skill set, let alone generalisation to a more distal skill-set such as emotion recognition. Perhaps this could be framed a little more speculatively.
d. It would be helpful in the Discussion to provide some consideration of the fact that correlations were not established between emotion recognition and age.

Additional comments

Grammar/copy-editing/expression clarity issues
1. A few instances of poor subject-verb agreement are noted in the paper, e.g., line 57 in the second page of the intro: population  has; in the Method, line 169 set  has.
2. On the same page, line 71 – consider substituting “different” for “contradictory”
3. On the third page of the intro, line 82, the word “emotional” is repeated.
4. In the Method, line 117 – it is usual to use words rather than numerals at the start of the sentence (depending of course on journal conventions). This occurs elsewhere in the MS also.
5. On the same page as point 4 above, line 121 – add “criteria” after “DLD”.
6. Second page of method, line 124 - I suggest adding “probable” before “intellectual disability” and in line 125, adding “below the mean” at the end of the sentence.
7. Same page as point 6 above, line 133 – remove “in” within brackets.
8. Still on the same page, line 139 – as a Latin phrase “a priori” would normally be italicised. This occurs elsewhere also.
9. Method (under heading sample size and power calculations) – line 146 – change “suggest” to “suggested” and remove “a” after “detect”.
10. Same page as above, line 156 – “z-scores” is normally all lower case.
11. Still on the same page, line 158 – change “test” to “tests.
12. In the Method, above the heading “Analysis plan” (line 185) – I am not a fan of “inputted” – could this be changed to “submitted” or similar? Also there is a missing “to” before “give” in this same sentence.
13. In Method, para above heading “Sampling weights and missing data”, line 190 – change “coverts” to converts.
14. Same page as above, line 198 – monolingual is one word (no hyphen)
15. Same page as above – some form of inclusion/includes/ included is used four times between lines 206 and 208.
16. On the next page, line 212 – change predict to predicted.
17. Same page as above, change “pervious” to previous.
18. Same page as above, line 229 – the singular form is criterion (“criteria” is plural).
19. On the page with the heading “Results” it is not clear why there is a heading immediately under this “Participants”
20. On this same page, line 239 – add “who” after “participants”.
21. Final page of Results, line 293 – change surprised to surprise.
22. Discussion, third page, line 335 – insert “task” after emotion labelling.
23. My personal preference is to use lower-case n not N when describing sample size, but the authors should check the journal requirements on that.
24. Where DSM is mentioned in the Discussion, this needs an edition and a reference, as well as a citation in the Reference List.

---

## Round 0.2 · accepted · Accept

Thank you for comprehensively addressing the reviewers' comments. I am delighted to be able to accept you paper for publication in PeerJ.